# Switch from Ibalizumab to Lenacapavir in a Rescue Regimen for a Heavily Treatment-Experienced (HTE) Patient with Multidrug-Resistant (MDR) HIV-1 Infection

**DOI:** 10.3390/ijms26188881

**Published:** 2025-09-12

**Authors:** Salvatore Martini, Lorenzo Salmoni, Roberta Palladino, Antonio Russo, Nunzia Cuomo, Adriana Raddi, Mario Starace, Carmine Minichini, Mariantonietta Pisaturo, Nicola Coppola

**Affiliations:** 1Department of Mental Health and Public Medicine, Section of Infectious Diseases, University of Campania Luigi Vanvitelli, 80138 Naples, Italy; lorenzo.salmoni@studenti.unicampania.it (L.S.); roberta.palladino@studenti.unicampania.it (R.P.); antoniorussoarar@gmail.com (A.R.); mariostarace1984@libero.it (M.S.); carmine.minichini@alice.it (C.M.); mariantonietta.pisaturo@unicampania.it (M.P.); nicola.coppola@unicampania.it (N.C.); 2U.O.C. Microbiologia e Virologia, Azienda Ospedaliera Specialistica dei Colli Monaldi-Cotugno-C.T.O., 80131 Napoli, Italy; nunzia.cuomo@ospedalideicolli.it (N.C.); adriana.raddi@ospedalideicolli.it (A.R.)

**Keywords:** HIV, HTE, IBA, switch, resistances, LEN

## Abstract

Despite the progress of antiretroviral therapy, there are still some patients with (MDR) HIV infection. In this case, international guidelines suggest using new-generation drugs, such as Ibalizumab (IBA) or Lenacapavir (LEN), in combination with an optimized background regimen. Our clinical case concerns a Heavily Treatment-Experienced (HTE) patient with MDR HIV-1 infection. Rescue therapy began in April 2022, combining residual drugs with low-level resistance and IBA. At this time, HIV-RNA results included 37.800 copies/mL, and CD4+ included 147 cells/µL. IBA was administered intravenously every 15 days. After 12 months, to optimize adherence, IBA was re-placed by LEN, which has long-acting posology, with subcutaneous injections every 6 months. IBA achieved viral suppression in only one month with an improvement in the CD4+ count and showed a progressive disappearance of viral mutations in the reservoir. It was well tolerated except for the onset of hypertension after infusions. After 12 months, IBA was switched to LEN, which showed good tolerability, preserving efficacy and stable pressure on HIV-DNA. Our case report about an HTE patient shows that IBA was efficacious in the rescue regimen, while also acting on the reservoir. LEN, adopted in a switch strategy which differed from that described in the literature, preserved efficacy and stable pressure on HIV-DNA.

## 1. Introduction

Advances in antiretroviral therapy have improved the efficacy of regimens that are characterized not only by the control of viral replication but also by the optimization of tolerability and adherence. This aspect improves quality of life, which is fundamental to ensuring the durability of treatment. However, there are some patients who present with multiple drug-resistant (MDR) viruses and need rescue therapies. The international guidelines in this case suggest using regimens based on new drugs, which have been recently discovered and characterized as a new target of treatment, associated with the other antiretrovirals with residual efficacy that represent the optimized background regimen (OBR) [1].

The conditions that may explain the evidence of patients with MDR HIV may be varied, and can be related to imperfect patient adherence, but also to the adoption of non-fully active regimens. This is more frequently observed for patients with an old infection, who were treated initially with regimens characterized by a low genetic barrier. In this context, regarding patients’ adherence, it was possible to observe treatment failure for the progressive selection of viral mutations. Before the advent of HAART, in fact, patients were treated with the first discovered antiretrovirals, such as nucleoside reverse transcriptase inhibitors (NRTIs), alone or in combination with another drug of the same class. This type of treatment was not very efficacious and, for this reason, when clinicians introduced a new drug in combination with NRTIs (the “backbone” of the HAART), the new regimen could show low durability for backbone corruption. The progressive increase in resistance mutations created the basis for other successive failures in these patients, creating a vicious cycle. Another possible eventuality for a patient is superinfection with another resistant HIV virus. This condition may cause the fusion of the two viruses with different patterns of resistance that may create a chimera, adding mutations to the two patients.

The effectiveness and high remission of the antiretrovirals in recent years have significantly reduced the proportion of patients experiencing viral failure and developing resistance. The same diffusion of regimens in a single-tablet regimen (STR) with good long-term tolerability considerably improved adherence. 

Achieving a high rate of viral suppression for HIV plays a major role in reducing the impact of the virus, leading to decreased transmission, morbidity, and mortality.

In December 2020, UNAIDS released a new set of ambitious targets for 2025:95% of all people living with HIV should know their HIV status.95% of all people diagnosed with HIV should receive sustained antiretroviral therapy (ART).95% of all people receiving ART should achieve viral suppression by 2025 [2].

This strategy aims to reduce the inequalities that fuel the AIDS epidemic and to place people at the center of the HIV response.

In 2023, an estimated 86% [69–98%] of people living with HIV were aware of their status, 77% [61–89%] were receiving treatment, and 72% [65–80%] were virally suppressed [3].

Focusing on the European region, in 2023, 93% of people living with HIV were receiving treatment, while 73% were virally suppressed [4]. At the same time, the prevalence of heavily treatment-experienced (HTE) people with HIV in Europe ranged from 1.9% to 10.4%, depending on the country and cohort analyzed [4].

In a EuroSIDA study, HTE individuals accounted for 10.4% of the 15,570 participants under follow-up between 2010 and 2016 [5].

In the OPERA cohort study in the United States, researchers enrolled 24,183 ART-experienced people living with HIV (PLWH), of whom 2277 (9%) met the criteria for HTE [6].

In the Italian ICONA cohort, among 8758 PLWH, 163 (1.9%) were identified as HTE [7].

The representation of HTE PLWH differs between the United States and Italy. These differences reflect varying historical patterns of treatment access, ART initiation timing, and surveillance systems. In Italy, data from the ICONA cohort show that only 1.9% of PLWH meet the criteria for HTE status, with 18% of this subgroup experiencing ongoing immuno-virological failure [7].

In the United States, the annual prevalence of HTE PLWH with Limited Treatment Options (LTOs) was 5.2–7.5% from 2000 to 2006, declined significantly to 1.8% in 2007, and fell below 1% from 2012 through 2017 [8].

Individuals classified as HTE often face challenges due to limited antiretroviral treatment options. 

Factors commonly associated with HTE status include:A low CD4 nadir at ART initiation.Co-infection with hepatitis C virus (HCV).A long treatment history.

The regimens available today are the result of decades of progress in efficacy, genetic barriers, forgiveness, pill burden, and tolerability. These advances have substantially improved patients’ long-term prognosis.

As a result, many patients living with HIV today were infected 20–30 years ago and have survived to witness the evolution of ART regimens, which have become better tolerated with a progressive reduction in the pill burden and the number of daily doses.

Patients who were exposed to older, more complex regimens, characterized by a high pill burden and more side effects, often struggled with adherence, despite the life-saving nature of the therapy.

In such conditions, some patients did not maintain good adherence, leading to therapeutic failure and the accumulation of resistance mutations.

Furthermore, therapy sequencing strategies at the time relied on the gradual introduction of new drugs to manage failures associated with low genetic barriers.

This approach led some patients to accumulate multiple resistance mutations, limiting their future options and resulting in HTE cases with multidrug-resistant (MDR) HIV.

Although rare, these patients present a significant clinical challenge, requiring physicians to find effective and preferably long-lasting solutions.

The development of rescue regimens has been made possible by the introduction of drugs with novel mechanisms of action.

These are used alongside residual antiretrovirals that still exhibit partial efficacy, enabling viable treatment options [9].

Currently available drugs for rescue regimens in clinical practice include

Ibalizumab (IBA)—a monoclonal antibody that inhibits the CD4-gp120 interaction [10].Fostemsavir—an entry inhibitor [11].Lenacapavir (LEN)—a capsid inhibitor [12].

In our clinical case, we initially adopted Ibalizumab (IBA) as the first phase of treatment, subsequently switching to Lenacapavir (LEN), which offered a more favorable dosing schedule.

The process by which HIV-1 enters susceptible target cells (primarily CD4+ lymphocytes) is a complex sequence of steps that includes viral attachment, co-receptor binding, and fusion [13,14].

Several HIV-1 entry inhibitors with different mechanisms of action have been developed to interact with specific steps in this process [14], thereby preventing the virus from entering and infecting immune cells [13].

IBA is a first-in-class CD4-directed post-attachment HIV-1 inhibitor and the first monoclonal antibody approved by the U.S. Food and Drug Administration (FDA) in March 2018 for the treatment of multidrug-resistant (MDR) HIV-1 infection. It is indicated in combination with other antiretroviral agents for heavily treatment-experienced (HTE) adults who are failing to their current therapy [15].

Treatment with monoclonal antibodies offers several advantages, including a novel mechanism of action, the potential to restore CD4+ T cell counts, minimal risk of acquired resistance, and lower toxicity compared to traditional antiretroviral drugs.

IBA binds to the extracellular domain of the CD4+ T cell receptor and inhibits HIV-1 entry by preventing conformational changes in the gp120-CD4 complex that are necessary for co-receptor binding and membrane fusion [16].

The efficacy of IBA in treating MDR HIV-1 infection was initially demonstrated in two randomized, double-blind Phase II trials: a 48-week, placebo-controlled Phase IIa study (TNX-355.03) [16] and a 24-week Phase IIb dose–response trial (TMB-202) [17].

Its therapeutic effectiveness was further confirmed in a multicenter, open-label Phase III trial (TMB-301) [18], which enrolled 40 adults (≥18 years) with MDR HIV-1 infection and a viral load of >1000 copies/mL.

The trial included

A control period (days 0–6), during which patients continued their failing ART regimen;A functional monotherapy period (days 7–13), during which participants received a 2000 mg intravenous loading dose of IBA;A maintenance period (day 14–week 25), in which patients started an optimized background regimen (OBR) and continued taking IBA at 800 mg intravenously every 14 days, beginning on day 21.

IBA demonstrated potent antiviral activity in this population [18]. A significantly higher number of patients achieved a viral load reduction of ≥0.5 log_10_ copies/mL after IBA was added to their failing regimen.

Preliminary real-world data further support the antiviral efficacy of IBA in MDR HIV-1 infection [19].

IBA was generally well tolerated. The most commonly reported adverse effects included diarrhea, dizziness, fatigue, nausea, and pyrexia [19].

The second innovative drug used in our patient was Lenacapavir (LEN), a novel, first-in-class, multistage, selective inhibitor of the HIV capsid protein. LEN interferes with multiple phases of the HIV replication cycle [12].

The development of LEN stems from recent advances in understanding the role of the HIV capsid. Once considered only a structural component, the capsid is now known to play a key role in nuclear entry, interaction with host factors, and integration.

By binding to two adjacent subunits of the HIV-1 capsid protein, LEN disrupts essential interactions for several phases of replication, including capsid-mediated nuclear import, virion production, and proper capsid core formation [20,21].

Viruses produced in the presence of LEN display abnormally shaped capsids. Although they can enter new target cells, they are unable to replicate.

Clinical studies have demonstrated that LEN achieves significant antiviral activity and maintains effective pharmacokinetic exposure for up to six months following a single subcutaneous injection.

Phase 1, 2, and 2/3 trials support the use of LEN in both HTE and treatment-naïve people with HIV (PWH), in combination with other antiretrovirals.

The CAPELLA international Phase 2/3 trial, conducted across 42 sites, enrolled 72 adults with MDR HIV-1.

After a two-week oral LEN loading phase, participants received subcutaneous LEN every 26 weeks with an OBR.

At Week 104:A total of 62% of participants had HIV-1 RNA < 50 copies/mL.Excluding missing data, 82% achieved viral suppression.The mean CD4+ cell count increased by 122 cells/μL.The proportion of participants with CD4+ < 200 cells/μL decreased from 64% to 29%.

LEN resistance emerged in 14 participants (20%), all of whom lacked fully active agents in the OBR or demonstrated poor adherence.

Seven of these individuals achieved re-suppression (HIV-1 RNA < 50 copies/mL) while continuing LEN.

No grade 4 or serious treatment-related adverse events were reported, and only one participant discontinued due to an injection site reaction [22,23].

The CAPELLA trial demonstrated significant virological success in patients with a virus resistant to at least two antiretroviral classes and a limited number of remaining treatment options.

LEN was generally well tolerated, with no treatment discontinuations due to drug-related adverse events.

The only discontinuation was due to a possible hypersensitivity reaction at the injection site, characterized by redness, edema, and mild pain.

These local reactions typically resolved within a few days and were not described as painful after the acute phase.

In some cases, red nodules appeared at the injection site and persisted for a few weeks.

It is important to note that, given the complex treatment histories and cumulative toxicities in this patient population, assessing tolerability is more challenging than in treatment-naïve individuals.

In contrast, in the CALIBRATE study, which enrolled treatment-naïve patients, tolerability assessment was more straightforward.

CALIBRATE is an open-label, Phase 2, active-controlled, induction-maintenance study involving treatment-naive PWH with CD4+ counts ≥ 200 cells/μL.

Participants were randomized into four treatment groups.

LEN, administered either subcutaneously or orally, in combination with emtricitabine/tenofovir alafenamide, resulted in

A viral suppression rate of 90% in group 1.A viral suppression rate of 85% in groups 2 and 3,A viral suppression rate of 92% in group 4.

At Week 54, the overall virological suppression was 90%.

No serious adverse events related to LEN were reported [23].

### Case Report

Our clinical case concerns a 70-year-old individual, who is a typical example of a person with a long-standing diagnosis of HIV infection and a 30-year history of antiretroviral therapy (ART). Over the course of his treatment, the patient underwent multiple regimen changes, guided by resistance testing in response to progressive therapeutic failures. One of the main challenges for patients with long-term HIV infection lies in the limitations of the early antiretroviral drugs initially used to suppress viral replication.

The first treatment regimens were largely based on monotherapy with nucleoside reverse transcriptase inhibitors (NRTIs), which, when used alone, were unable to achieve durable viral suppression. This patient was diagnosed in 1995, a time when protease inhibitors (PIs) had just been discovered and were not yet available for clinical use. As a result, over time, the patient accumulated resistance mutations, particularly in the NRTI class, which compromised the efficacy of later triple-drug regimens. These regimens were administered to a patient with a pre-existing, weakened NRTI backbone, which was insufficient to support or protect the third drug effectively. This created a vicious cycle, in which initial virologic failures increased the likelihood of subsequent failures, especially in the context of antiretroviral agents that, at the time, had lower potency and genetic barriers compared to current treatments. Such factors, including cross-resistance, likely contributed to the development of a multidrug-resistant (MDR) HIV strain in many long-term patients, as occurred in the individual described in this case. In this patient’s therapeutic history, the regimens prescribed over the decades helped manage progressive virologic failures but were unable to prevent the accumulation of resistance. In recent years, the patient was treated with a rescue regimen including drugs that retained partial activity, based on results from resistance testing. These drugs, although now largely outdated, were still occasionally useful. At one point, newer agents were introduced, but the patient experienced further virologic failure. It became necessary to construct a new rescue regimen combining an optimized background regimen (OBR) with Ibalizumab (IBA), a recently available drug with a novel mechanism of action. This therapeutic strategy proved effective, and the patient continued the regimen for approximately 12 months. IBA, a monoclonal antibody that blocks the fusion of HIV with CD4+ lymphocytes, was administered intravenously every two weeks. However, this administration schedule was challenging for long-term adherence [9]. Therefore, to improve treatment convenience and maintain virologic suppression, IBA was replaced with Lenacapavir (LEN). Although LEN is typically used in rescue regimens for HTE patients with active viral replication, in this case, it was used in a switch strategy, which was aimed at preserving efficacy while improving the patient’s quality of life and ensuring the durability of the treatment. LEN is in fact administered subcutaneously once every six months, offering a more manageable dosing schedule [12].

## 2. Study Design

We analyzed the clinical case of a heavily treatment-experienced (HTE) patient to assess the outcomes of a rescue regimen based on IBA + OBR, and to evaluate a switch strategy to a LEN-based regimen to improve treatment adherence.

We monitored

HIV viral load evolution.CD4+ T-cell count.HIV-DNA levels.

These parameters allowed us to evaluate not only the plasma virologic efficacy of the new drugs, but also their impact on the viral reservoir during follow-up.

In addition, we assessed the long-term tolerability of both rescue regimens.

## 3. Result

This case report focuses on a patient who has been followed at our clinic for over 30 years.

He is currently 70 years old and worked as a chef on offshore platforms.

He is infected with HIV-1 subtype B, classified as CDC A3, and reported a history of unprotected intercourse with same-sex partners, recognized as a risk factor for HIV infection (see Table 1).

Over the years, the patient tried numerous treatment regimens, progressively developing resistance mutations across all antiretroviral classes, before eventually meeting the criteria for HTE status and harboring an MDR HIV strain.

All regimens were selected based on genotypic resistance test results. In some in-stances, older drugs with residual antiviral activity were reintroduced based on sus-ceptibility profiles.

We summarized the patient’s treatment history and corresponding resistance test results in Table 2.

We also reported two HIV genotypic resistance tests with corresponding Stanford HIVdb interpretations.

The first test, conducted in 2013, showed the emergence of resistance mutations to integrase strand transfer inhibitors (INSTIs). These mutations were not confirmed in subsequent resistance tests (see Figure 1).

The second test was performed in January 2022, prior to the introduction of Ibalizumab (IBA) into the patient’s treatment regimen (Figure 1).

In this case, the Genotypic Sensitivity Score (GSS) was as follows:NRTIs: 0.75 (indicating potential low-level resistance).NNRTIs: 0 (indicating high-level resistance).PIs: 0.03 (indicating high-level resistance).INSTIs (RNA): no resistance mutations detected.

A subsequent proviral DNA resistance test, performed in April 2022, revealed archived INSTI resistance mutations, with intermediate resistance to dolutegravir (DTG) and bictegravir (BIC) (Figure 1).

These findings clearly indicated that the optimized background regimen (OBR) alone was insufficient to achieve sustained viral suppression.

Consequently, it became necessary to start a rescue regimen including next-generation antiretroviral agents.

At baseline, in April 2022, prior to initiating the rescue therapy, the patient presented with a CD4+ T-cell count of 147 cells/μL and an HIV-1 viral load of 31,700 copies/mL.

At that time, the patient was receiving a regimen consisting of Tenofovir disoproxil fumarate/Emtricitabine + Etravirine + Dolutegravir (TDF/FTC + ETV + DTG), selected based on prior HIV resistance testing.

Following a new episode of viral rebound, a subsequent resistance test was performed. The results revealed the presence of resistance mutations across all antiretroviral classes, except for integrase strand transfer inhibitors (INSTIs).

These findings made it necessary to initiate a rescue regimen with next-generation antiretroviral agents (Figure 2).

The rescue protocol involved discontinuation of the failing antiretroviral therapy, followed by initiation of Ibalizumab (IBA), the first monoclonal antibody developed to block the interaction between the HIV virus and CD4+ lymphocytes [15].

IBA was administered initially as functional monotherapy for 7 days, then combined with an optimized background regimen (OBR) composed of drugs with residual antiviral activity, selected based on genotypic resistance testing.

The OBR in this case included Tenofovir disoproxil fumarate/Emtricitabine + Etravirine + Dolutegravir (TDF/FTC + ETV + DTG), used in combination with IBA.

This approach led to a 2-log reduction in viral load within the first 7 days of IBA monotherapy, followed by the introduction of the OBR, which allowed the patient to achieve full viral suppression within two additional weeks (see Figure 2).

Typical functional monotherapy phases include continuation of ineffective background drugs, which is functionally analogous to the true monotherapy we have adopted with IBA. In our case, the patient had already been receiving the same OBR prior to IBA initiation.

To optimize the efficacy of the OBR, we decided to temporarily suspend it during the IBA loading dose phase and reintroduce it afterward, as per protocol. This strategy aimed to maximize the OBR’s virologic impact after the initial viral load reduction achieved by the monoclonal antibody.

IBA was well tolerated, with the only adverse event being a mild increase in blood pressure observed after infusion.

IBA is administered via intravenous infusion every 15 days.

Regarding efficacy, the patient not only achieved virologic suppression but also experienced a significant recovery in CD4+ T-cell count, which had never previously exceeded 200 cells/μL prior to this rescue regimen (see Figure 2).

Another notable finding was the depth of virologic suppression, evidenced by the progressive disappearance of archived resistance mutations detectable in HIV-DNA over time.

This observation does not imply a reversal of resistance, but rather suggests that deep viral suppression reduced the amount of amplifiable virus in the latent reservoir, making archived mutations undetectable (see Table 3).

Specifically

After 1 month, INSTI resistance mutations were no longer detectable in proviral DNA.After 3 months, NNRTI mutations had also disappeared, along with a reduction in previously detected PI and NRTI mutations.

In April 2023, the patient discontinued Ibalizumab (IBA) and continued therapy with the OBR alone.

Although plasma viral suppression was partially maintained, the pressure on the viral reservoir was rapidly reduced, as evidenced by the early reappearance of archived resistance mutations.

This clinical observation underscores the crucial role of new-generation antiretroviral agents, such as IBA, in combination with OBR, to achieve sustained long-term virologic efficacy, particularly in heavily treatment-experienced patients (see Figure 3 and Table 3).

In May 2023, a new resistance test was performed, which confirmed the reemergence of previously archived mutations to protease inhibitors (PIs), non-nucleoside reverse transcriptase inhibitors (NNRTIs), and nucleoside reverse transcriptase inhibitors (NRTIs) (Table 2). As a result, the clinical team decided to introduce Lenacapavir (LEN) to replace IBA, taking advantage of LEN’s more convenient dosing schedule, which comprises a single subcutaneous injection every six months. This strategy was aimed at improving adherence and overall quality of life, offering a more comfortable long-acting alternative to IBA. The LEN regimen began with an oral loading phase:The patient received 600 mg on Day 1 and Day 2.The patient received 300 mg on Day 8.

On Day 15, the patient received two subcutaneous injections of 463.5 mg, with the same dose repeated every six months thereafter. LEN was well tolerated, with only the appearance of indolent, subcutaneous weals at the injection site. These were painless, showed no signs of inflammation, and although palpable for months, progressively reduced in size over time. In terms of efficacy, HIV-RNA remained suppressed, and the CD4+ count continued to increase (see Figure 2). Interestingly, LEN was used here in a non-conventional context. It was not introduced in a patient with ongoing viral replication and high-level resistance, but rather as part of a switch strategy after prior successful suppression with a rescue regimen (IBA + OBR). The aim was to maintain virologic control while improving tolerability and adherence, leveraging LEN’s pharmacokinetic advantage of biannual dosing. In this context, without LEN, the OBR alone would likely have failed, as the included drugs retained limited activity. LEN ensured the durability and stability of the regimen, not only by maintaining plasma viral suppression, but also by enhancing the clearance of archived resistance mutations from the viral reservoir, as shown by HIV-DNA analyses:After 1 month, archived PI mutations disappeared, with a reduction in NRTI and NNRTI mutations (Table 3).At 6 months, NNRTI mutations were no longer detectable, and only a single mutation remained in each of the PI, NRTI, and INSTI classes.After 12 months, the impact on the reservoir was comparable to that seen with IBA, with no detectable HIV-DNA mutations (see Figure 3).

## 4. Methods

We collected samples every 30 days following initiation of the rescue regimen. The following were assessed:Clinical examination.Biochemical parameters.CD4+ T-cell count.HIV-RNA (plasma viral load).HIV-DNA quantification and amplification to detect resistance mutations in the viral reservoir.

All procedures were conducted in accordance with the Ethics Committee of the University of Campania “L. Vanvitelli”, Naples, and complied with the Declaration of Helsinki (1964) and its later amendments. Informed consent was obtained from the patient.

### 4.1. Serological and Molecular Analysis

Plasma HIV-1 viral load was measured using real-time PCR, with a detection limit of 40 copies/mL.

For resistance testing

Viral RNA was isolated from 200 μL of serum using the QIAamp RNA viral kit (Qiagen GmbH, Hilden, Germany).Genotyping was performed using a validated in-house methodology developed by the ANRS AC11 Resistance Study Group, including PCR and Sanger sequencing of the pol gene, covering:○Protease (PR).○Reverse transcriptase (RT).○Integrase (IN).

Resistance interpretation was conducted using

The IAS-USA mutation list.The Stanford HIV Drug Resistance Database (https://hivdb.stanford.edu, accessed on 21 August 2025).The Stanford CPR algorithm for primary resistance analysis (https://hivdb.stanford.edu/cpr/, accessed on 21 August 2025).

It is worth noting that the literature suggests that Sanger sequencing identifies viruses susceptible to key drugs in 88% of patients, while ultra-deep sequencing (UDS) detects susceptibility in 80% [24].

Resistance testing was performed for all patients with viral loads of >1000 copies/mL.

### 4.2. HIV-DNA Quantification and Resistance Testing

HIV-DNA was analyzed using the DIATHEVA HIV-1 DNA Test PRO, a quantitative PCR assay that detects and quantifies all forms of intracellular HIV-1 DNA via fluorescent probe-based amplification.

Key features of the assay:Duplex PCR targeting both HIV-1 DNA and human telomerase reverse transcriptase (hTERT) for relative quantification and internal control.Robust performance in the presence of PCR inhibitors.Room temperature setup.Standard curve included with five levels of HIV-1 copy numbers and cell content.

### 4.3. Flow Cytometry

Lymphocyte subsets (CD4+, CD8+) were measured using flow cytometry with monoclonal antibodies and a fluorescence-activated cell sorter (FACS) scan (Becton Dickinson, Mountain View, CA, USA).

Routine biochemical and hematologic analyses were performed using standard laboratory protocols.

### 4.4. Quality of Life Assessment

Quality of life was not assessed with standardized or validated tools.

Instead, subjective improvement was documented anamnestically, based on the patient’s self-reported experience regarding adherence, tolerability, and convenience.

## 5. Discussion

People living with HIV (PWH) who have limited treatment options due to extensive drug resistance are often classified as difficult to treat (DTT). In the ICONA cohort, DTT individuals represent approximately 6.5% of the total cohort and constitute a particularly vulnerable subgroup with an increased risk of virological failure [7].

The prevalence of multidrug-resistant (MDR) HIV, defined by resistance to three or more antiretroviral classes, showed a significant decline between 2000 and 2010, dropping from 26% to 6%, largely due to the improved efficacy of antiretroviral therapy (ART). However, since 2010, this percentage has plateaued, stabilizing at around 5%. European cohort data corroborate this, with three-class resistance reported in 6.9% of treated individuals [5].

As a result, managing heavily treatment-experienced (HTE) patients remains a substantial clinical challenge. The construction of rescue regimens requires careful consideration, not only of virological efficacy, but also of factors like pill burden and treatment adherence. Current international guidelines recommend combining an optimized background regimen (OBR) with a drug possessing a novel mechanism of action, to which the virus has not been previously exposed [1].

This recommended strategy was successfully applied in the clinical case presented. However, what distinguishes this case report are two particularly novel elements that are rarely discussed in the literature.

### 5.1. Switch from IBA to LEN to Optimize Durability and Adherence

The first novel aspect of this case is the strategic switch from Ibalizumab (IBA) to Lenacapavir (LEN), both innovative agents with unique mechanisms of action, in the context of treatment maintenance rather than failure.

IBA is a monoclonal antibody targeting the CD4 receptor, which is administered intravenously every two weeks. While effective, its frequent infusion schedule can compromise long-term adherence and affect patient quality of life. LEN, in contrast, is a first-in-class capsid inhibitor with long-acting pharmacokinetics, allowing for subcutaneous administration every six months. This significant difference in posology presents an opportunity to optimize adherence, particularly in older patients or those with a long history of ART exposure.

In our case, LEN was introduced not in the typical scenario of ongoing viral replication, but as part of a switch strategy in a virologically suppressed patient who had previously responded to IBA-based therapy. The goal was to maintain virological suppression while improving treatment tolerability, convenience, and quality of life.

LEN was well tolerated, with only minor injection-site reactions, and it maintained plasma viral suppression while supporting a continued increase in CD4+ count. Its oral lead-in phase, followed by twice-yearly subcutaneous injections, could represent a paradigm shift in HIV management, transitioning from daily oral therapy to ultra-long-acting regimens, which may help overcome the psychological burden of daily medication. This could provide a new sense of “freedom from therapy”, particularly in aging or treatment-fatigued populations.

Moreover, this case suggests that LEN could have a broader role in future HIV strategies, not only in rescue regimens for HTE patients, but also potentially in maintenance therapy and pre-exposure prophylaxis (PrEP).

### 5.2. HIV-DNA Reservoir Monitoring and Resistance Evolution

The second innovative aspect of this case was the longitudinal monitoring of HIV-DNA in peripheral blood mononuclear cells (PBMCs), allowing for an evaluation of the reservoir dynamics and archived resistance mutations, independent of plasma viral suppression.

Routine HIV-RNA measurements reflect circulating virus, but they do not capture the latent reservoir, which continues to pose a barrier to cure. In this case, serial HIV-DNA testing provided insight into the depth of antiviral activity during both the IBA-based and LEN-based phases of treatment.

Interestingly, while total HIV-DNA copy numbers remained stable, we observed a progressive disappearance of archived resistance mutations during follow-up. This was particularly notable following the initiation of IBA, and the trend continued after the switch to LEN. Mutations affecting PI, NNRTI, NRTI, and even INSTI classes that were initially detectable became undetectable over time (Table 3, Figure 3).

This phenomenon may be interpreted not as a reversal of resistance, but rather as a reduction in the pool of amplifiable proviral DNA, reflecting a deep suppression of replication-competent virus. Both IBA and LEN may have contributed to this reservoir clearance effect, either directly or indirectly by maintaining potent, durable suppression in the absence of ongoing selective pressure from ineffective drugs.

### 5.3. HIV-DNA Resistance Analysis and CD4+ Recovery: Markers of Deep Treatment Efficacy

In our case, HIV-DNA resistance testing was performed not due to virological failure, but as a research tool to assess the depth of antiretroviral efficacy beyond the detection threshold of plasma viral load assays. The patient maintained undetectable plasma viremia with salvage therapy, and no residual viremia was detectable. Therefore, the observed changes in proviral DNA reflect true suppression of viral reservoirs, not just plasma control. These findings are consistent with prior research, notably by Nouchi Et Al. [24], who demonstrated that archived drug resistance-associated mutations (DRAMs) in cell-associated HIV-1 DNA, particularly against reverse transcriptase inhibitors, can disappear over time in patients under long-standing virological suppression. Their study suggests that the lack of selective pressure may facilitate the natural decay or outgrowth of resistant viral clones, contributing to reservoir remodeling. In our case, the gradual disappearance of resistance mutations in HIV-DNA (Figure 3) aligns with the sustained efficacy of the salvage regimens—first IBA + OBR, and later LEN + OBR. Notably, when IBA was discontinued and the patient continued OBR alone, archived resistance mutations that had previously become undetectable reappeared in the proviral DNA. This reemergence occurred despite maintained plasma suppression, underscoring that reservoir dynamics can precede clinical failure and highlighting the importance of fully effective combinations to control both plasma viremia and proviral replication. If LEN had not been introduced at that point, the reappearance of DRAMs may have predisposed the patient to future virological failure. The subsequent re-suppression and disappearance of these mutations following LEN administration further supports its deep antiviral activity, even in the context of stable plasma suppression. Another striking observation was the progressive and sustained increase in CD4+ T-cell count following the introduction of the rescue regimens. Prior to IBA initiation, despite various regimens achieving viral suppression, the patient’s CD4+ count remained consistently below 200 cells/µL. This plateaued immunological recovery is a known challenge in treatment-experienced individuals. However, after starting IBA and continuing through the switch to LEN, the patient’s CD4+ count rose significantly, surpassing 200 cells/µL for the first time in years and reaching a net gain of 130 cells/µL (Figure 2). This suggests that IBA and LEN may have additional immunological benefits beyond viral suppression. The mechanism behind this improvement is not fully understood, as CD4+ recovery is often poorly correlated with viral load dynamics in long-term-treated individuals. Nonetheless, this observation supports a potential indirect immunomodulatory role of these new agents and offers a promising avenue for improving clinical outcomes in patients with low CD4+ nadirs. Importantly, a CD4+ count below 200 cells/µL remains a strong predictor of morbidity and mortality, making this improvement clinically significant. To our knowledge, there are few published data on the switch from IBA to LEN in real-world settings. Our findings are in accordance with a recent Italian case report published in 2024, where authors, using data from the PRESTIGIO cohort, described a successful LEN-based switch strategy in a younger HTE patient previously treated with IBA [25]. These cases highlight the potential of new-generation antiretroviral agents with novel mechanisms of action to transform the management of HTE patients. The strategic deployment of IBA and LEN, not only to manage virological failure but also to improve adherence, reservoir control, and immune restoration, represents an important advancement in the field. However, further real-world data and longitudinal studies are needed to fully characterize the pharmacodynamic and immunological profiles of these agents. Our case contributes to this evolving narrative by offering detailed insights into virological and immunological evolution, including HIV-DNA dynamics, in response to cutting-edge therapeutic strategies.

## 6. Conclusions

The efficacy of Ibalizumab (IBA) and Lenacapavir (LEN) in rescue regimens for heavily treatment-experienced (HTE) patients with multidrug-resistant (MDR) HIV-1 has been demonstrated in the literature [17,19,22,23]. However, our clinical case presents several unique aspects that may help shape future therapeutic strategies in this challenging population. This report highlights the potential of a sequential approach using novel agents, initially with IBA, followed by a strategic switch to LEN, not in the context of virological failure, but to optimize treatment tolerability, adherence, and long-term durability. The switch to LEN preserved virological suppression while significantly improving the quality of life through a more convenient dosing schedule (biannual subcutaneous injections), an important factor in maintaining adherence in aging, long-term-treated patients. In addition to robust plasma virological control, our case underscores the deep antiviral activity of these regimens at the reservoir level. Serial analyses of HIV-DNA mutations revealed a progressive decline and eventual disappearance of archived resistance mutations, suggesting a potential impact of these agents on the latent reservoir. This may have important long-term implications, not only for virological durability, but also for reducing chronic immune activation and residual inflammation, which are key contributors to comorbidity in people living with HIV. Furthermore, the significant increase in CD4+ T-cell count, achieved only after initiating these newer agents, may offer additional prognostic benefit, particularly in patients with historically poor immunological recovery despite virological suppression. In conclusion, this case supports the use of IBA and LEN not only as rescue options in MDR HIV-1 infection, but also as tools to refine and maintain effective therapy in complex HTE patients. Larger studies with extended follow-up are warranted to validate these findings and to explore the full clinical and immunological potential of new-generation antiretroviral agents in similar patient populations.

## Figures and Tables

**Figure 1 ijms-26-08881-f001:**
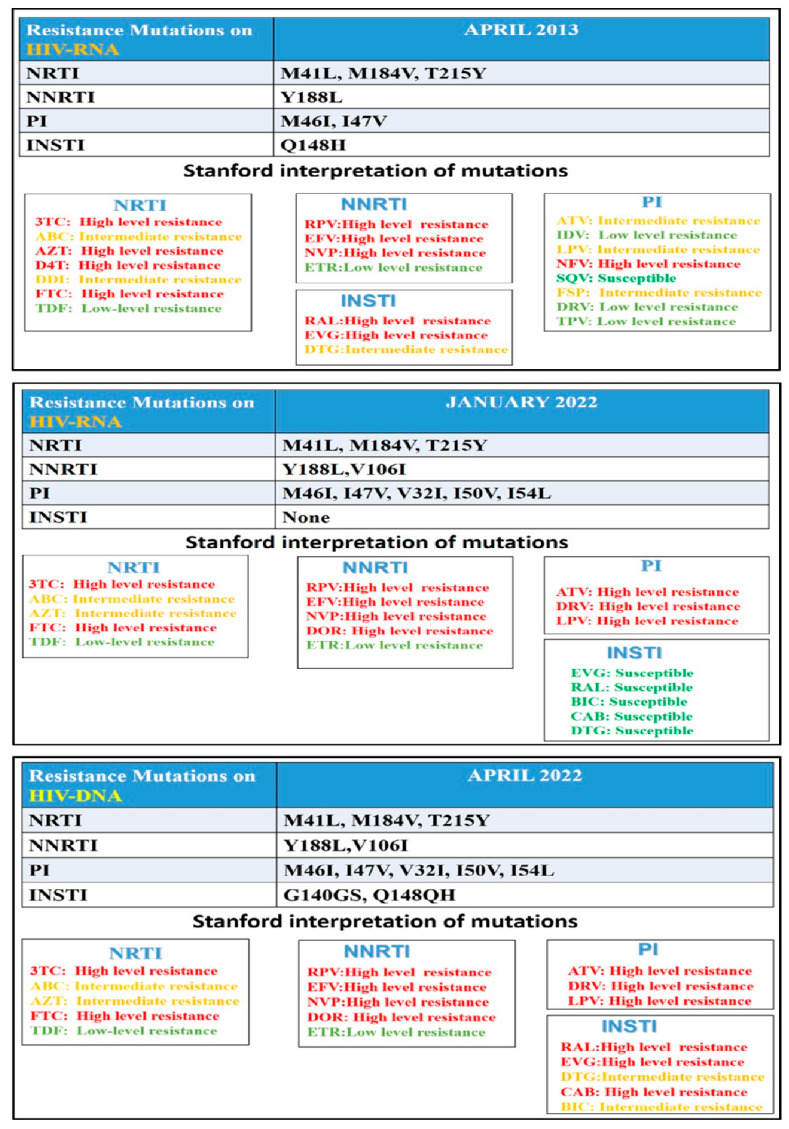
HIV resistance test with first onset of resistances for INSTI (April 2013); HIV resistance test before introducing IBA (January 2022); HIV resistance test performed on HIV-DNA (April 2022). AZT—Zidovudine; 3TC—Lamivudine; SQV—Saquinavir; NFV—Nelfinavir; DDI—Didanosine; D4T—Stavudine; IDV—Indinavir; TDF—Tenofovir disoproxil; NVP—Nevirapine; EFV—Efavirenz; ABC—Abacavir; FSP—Fosamprenavir; LPV—Lopinavir; ATV—Atazanavir; RAL—Raltegravir; DRV—Darunavir; FTC—Emtricitabine; TPV—Tipranavir; ETR—Etravirine; DOR—Doravirine; RPV—Rilpivirine; DTG—Dolutegravir; EVG—Elvitegravir; BIC—Bictegravir; CAB—Cabotegravir.

**Figure 2 ijms-26-08881-f002:**
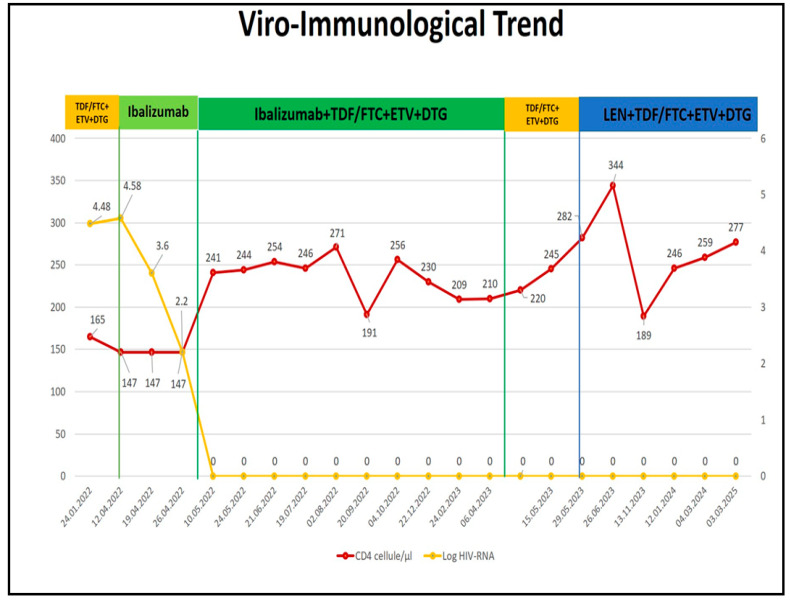
Viro-immunological data related to different rescue regimens. TDF, Tenofovir disoproxil; FTC—Emtricitabine; ETV—Etravirine; DTG—Dolutegravir; LEN—Lenacapavir.

**Figure 3 ijms-26-08881-f003:**
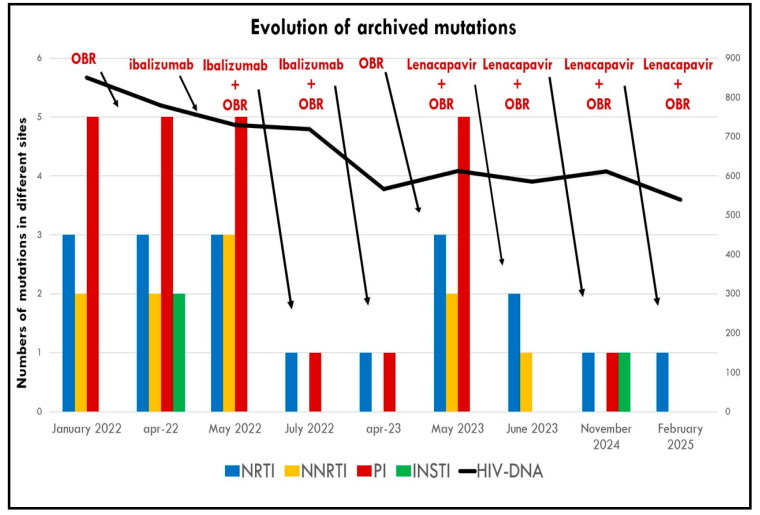
Kinetics of viral mutations evaluated for single class of regimens. OBR—optimized background regimen. The arrows clarify the effect on mutations induced by the previously introduced regimen.

**Table 1 ijms-26-08881-t001:** Anamnestic data of patient.

Patient Peculiarities	
Sex	Male
Risk factor	Reported occasional unprotected sexual intercourse with same sex partner
Year of diagnosis	1998
Age	70
CDC	A3
HIV subtype	B
Nadir CD4+	225
Number of therapeutic lines	15
Comorbidities	Dislipidemia, Hypertension

**Table 2 ijms-26-08881-t002:** Treatment history of patient with related available HIV resistance tests.

Resistance Mutations on HIV-RNA	98	99	00	03	04	05	07	08	09	10	11	12	13	14	18	21	22
REGIMEN	AZT 3TC SQV/r	AZT 3TC NFV	DDI IDV/r	TDF D4T NVP	TDF 3TC NVP	ABC 3TC FSP/r	ABC 3TC SQV/r	TDF 3TC LPV/r	RAL FTC DRV/r	ABC 3TC RAL FSP/r	TDF FTC RAL TPV/r	MVC DRV/r	RAL DRV/r	TDF ETV SQV/r ENF	TDF FTC DOR DRV/c	TDF ETV SQV/r	TDF FTC ETV DTG
NRTI					L210F					M41L, M184V, L210F, T215Y	M41L, M184V, T215Y		M41L, M184V, T215Y				
NNRTI					No Mutations					Y188L	Y188L		Y188L				
PI					Minor Mutations (L10V, L63P)					V32I, L33FL, M46I, I47V, I50V, F53FL, I54L	M46I, I47V		M46I, I47V				
INSTI					Not available					Not available	Not available		Q148H				
VIRAL TROPISM										R5			X4/DM				

NRTI—nucleoside reverse transcriptase inhibitors; NNRTI—non-nucleoside reverse transcriptase inhibitors; PI—protease inhibitors; INSTI—integrase strand transfer inhibitors; AZT—Zidovudine; 3TC—Lamivudine; SQV/r Saquinavir/ritonavir; NFV—Nelfinavir; DDI—Didanosine; IDV/r—Indinavir/ritonavir; TDF—Tenofovir disoproxil; D4T—Stavudine; NVP—Nevirapine; ABC—Abacavir; FSP/r—Fosamprenavir/ritonavir; LPV/r—Lopinavir/ritonavir; RAL—Raltegravir; DRV/r—Darunavir/ritonavir; FTC—Emtricitabine; TPV/r—Tipranavir/ritonavir; MVC—Maraviroc; ETV—Etravirine; ENF—Enfuvirtide; DOR—Doravirine; DRV/c—Darunavir/cobicistat; DTG—Dolutegravir.

**Table 3 ijms-26-08881-t003:** Viral mutations during salvage regimen.

Resistance Mutations	01/22 on HIV-RNA	04/22 on HIV-DNA	05/22 on HIV-DNA	07/22 on HIV-DNA	04/23 on HIV-DNA	05/23 on HIV-DNA	06/23 on HIV-DNA	11/24 on HIV-DNA	02/25 on HIV-DNA
REGIMEN	TDF/FTC +ETV +DGT	IBA	TDF/FTC +ETV +DGT +IBA	TDF/FTC +ETV +DGT +IBA	TDF/FTC +ETV +DGT	TDF/FTC +ETV +DGT +LEN	TDF/FTC +ETV +DGT +LEN	TDF/FTC +ETV +DGT +LEN	TDF/FTC +ETV +DGT +LEN
NRTI	M41L, M184V, T215Y	M41L, M184V, T215Y	M41L, M184V, T215Y	M184MV	M184MV	M41L, M184V, T215Y	M184MV, T215TI	K70KR	K70KR
NNRTI	Y188L, V106I	Y188L, V106I	Y188L, V106I, G190GE	None	None	Y188L, V106I	Y188YH	None	None
PI	M46I, I47V, V32I, I50V, I54L	M46I, I47V, V32I, I50V, I54L	M46I, I47V, V32I, I50V, I54L	L90LM	L90LM	M46I, I47V, V32I, I50V, I54L	None	L90LM	None
INSTI	None	G140GS, Q148QH	None	None	None	None	None	G140GS	None

## Data Availability

Restrictions apply to the availability of these data. Data were obtained from nicola.coppola@unicampania.it and are available with the permission of nicola.coppola@unicampania.it.

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
