# Peer review of "Switch from Ibalizumab to Lenacapavir in a Rescue Regimen for a Heavily Treatment-Experienced (HTE) Patient with Multidrug-Resistant (MDR) HIV-1 Infection"

_ijms, 2025, doi:10.3390/ijms26188881_

Round 1
Reviewer 1 Report
Comments and Suggestions for Authors
This is a case report of a highly treatment experienced person with HIV who achieved viral suppression with ibalizumab and maintained it after switching to lenacapavir despite low-level resistance to the background regimen.
The introduction introduces well the characteristics of long-term HIV survivors. Complete treatment history is missing. Interpretation and conclusion are good.
Some changes must be made:
The authors seem to suggest that viral suppression was achieved thanks to ibalizumab use. This is likely not true (please see below my comment about the GSS score). The erroneous claim must be corrected throughout, most importantly in the abstract.
Page 3: “mirror of their previous mistakes” is absolutely unacceptable. This is an offensive statement blaming people with HIV. It must be removed.
In the introduction, everything about PrEP should be removed as it has nothing to do with the case.
Same for statements at the end of the discussion. Please remove reference to PrEP, which has nothing to do with the current case. (Page 11: “About PrEP…” to “in practical terms”)
The entire section from page 6 “In every case,…” up to page 7 “becomes more careful and adherent” must be removed. It is at best a repeat of the introduction, and it must not be included in the “case report” section.
Instead, the treatment history of the case should be included with specific drugs used duration and resistance mutations (when available).
Case report page 6 mentions a “60-year-old [person]” (not “patient”, please edit). Whereas the case description page 8 indicates that this person is 70-year-old. Please edit.
Please provide the GSS score for the OBR used. This is essential to interpret the results. Without this information, statement regarding the success of LEN cannot be made (page 9). Notably, a quick look at Stanford suggests no resistance against the NRTI combination and low-level resistance against etravirine. There is also no baseline resistance against dolutegravir. Accordingly, it is likely that the person would have achieved and maintained viral suppression even in the absence of IBA. Please also discuss.
The authors must justify the choice of using IBA as monotherapy for 2 weeksbefore adding the OBR; this is highly unusual.
The report of disappearance of the resistance mutations in the proviral DNA may be one of the interesting observations from this paper. However, clearance of resistance mutations by class have been studied before. Please refer to these studies and discuss your results in this context. In addition, can the authors report their target-not-detected/target detected results for this person? There exists a possibility that the person is not fully suppressed with their OBR; thus, the authors should discuss the possibility that they are not evaluating the “reservoirs” but rather viral replication in cryptic sites or below the lower limit of detection of their assays. This must be discussed and mention of action on the reservoir must be removed.
Details about the DNA sequencing assay must be included in the Methods. Was this NGS, Sanger? Has the method been published prior?
I have a few minor additional comments:
Throughout the manuscript, please use first-person language that supports a respectful and non-discriminatory way of talking about a person with HIV or others engaged in care.
In the introduction, “destined to fail” is a too strong statement, please adjust.
Can the authors comment about the large difference in representation of HTE in the HIV population comparing the US and Italy?
In the introduction, page 3; please edit the possibly AI-generated statements: “Lower CD4 counts...” and “Co-infection with HCV…”, “multiple prior…”
Page 3: “collateral” ïƒ side effects
Page 6: this section should be removed anyway but “incredible” is not a scientific term to include in a publication
“The problem with patients…” please edit; this is not an acceptable start of a sentence plus the following statement concerns only those with resistance issues.
In study design: “to evaluate… and the evaluation” please edit.
Manufacturer’s information for the DNA Test PRO should be provided.
Page 8: “His risk factor” is not first-person language appropriate. Please edit for something like: “He reported engaging in unprotected intercourse with same-sex partners, which is a known risk factor for HIV infection”
Same consideration for Table 1 please “homosexual relationships”.
Page 9: “the pressure on reservoir”: unclear please edit and explain.
Page 9: “In this case, the drug has been”: very awkward, unclear statement; please edit.
Some of the discussion repeats the introduction and should be removed to streamline the discussion. The advantageous “posology” and potential benefits in terms of quality of life of LEN are mentioned several times (3 or 4 times) throughout the manuscript. Please spare the reader and streamline the text for readability.
Page 10: mentioning adherence improvement between IBA and LEN does not make sense are both are administered.
Page 10: “On the other hand…”: stating that ART treatment is unrelated to CD4 counts is incorrect. Please edit.
Page 11: “sequencing of [drugs]” does not exist. I assume the authors mean “sequential use of […] drugs. Please edit.
Page 9: “that highlighted mutations for all classes of drugs” is erroneous; no integrase mutations could be detected. Please edit.
Comments on the Quality of English LanguageThe manuscript is understandable but the language must be improved. I have suggested edits for some difficult statements but thorough editing would be useful. Some entire sections of the manuscript have to be removed.
Author Response
ANSWERS TO THE COMMENTS OF THE REVIEWER 1
Point 1: The authors seem to suggest that viral suppression was achieved thanks to ibalizumab use. This is likely
not true (please see below my comment about the GSS score). The erroneous claim must be corrected throughout,
most importantly in the abstract.
Answer to point 1: We thank the reviewer for his reflection on the role of ibalizumab in the treatment of
the patient in this case report. Indeed, there may be some doubts about this, and for this reason we have
included further clarifications in the text and figures that better define the role of drugs in this context. The
hypothesis that the efficacy of the rescue regimen is linked to the use of ibalizumab is supported by
observational and laboratory data, otherwise the whole clinical case would not have had any reason to be
described. First, when ibalizumab was introduced, the patient had a viral load of 4.5 log while being
treated only with intermediate-efficacy drugs (the same then included in the OBR associated to IBA),
which effectively cannot control viral replication. For this reason, we have modified Figure 1 adding
regimen used before IBA, to make the graph clearer. The same analysis of mutations in HIV RNA and
HIV DNA showed the presence of resistance to all drug classes, thus the regimen in place before the
introduction of ibalizumab had limited efficacy, confirmed by persistent viral replication. The
introduction of ibalizumab made a difference, as shown in the graph of Figure 1 and in Table 2 of
resistances that highlights the subsequent disappearance of mutations in the reservoir. Further evidence
is the fact that, during follow-up, upon discontinuation of ibalizumab, the patient, treated for a little
period with OBR alone, showed again mutations in the reservoir that had previously lost in HIV-DNA
analysis.
Point 2: Page 3: “mirror of their previous mistakes” is absolutely unacceptable. This is an offensive
statement blaming people with HIV. It must be removed.
Answer to point 2: Thank you for your comment on the sentence. I'm sorry it caused a misunderstanding,
as there was no intent to offend the patient. The sentence tried to highlights the concept that when a
patient develops resistances for some drugs, these mutations are archived and may cross-react with other
drugs of the same family. So whatever is compromised along the way then outlines the options still
available today. In every case, as suggested, we corrected the text to avoid any misinterpretations.
Point 3: In the introduction, everything about PrEP should be removed as it has nothing to do with
the case.
Answer to point 3: Thank you for the suggestion. The references to the PREP programs for lenacapavir
served to highlight the new studies on the once-a-year dosage with pharmacokinetic data even superior
to the current ones every 6 months and this could be an added value for the drug in prospective therapy
and not only in PREP. We have removed the references to PREP in the text.
Point 4: Same for statements at the end of the discussion. Please remove reference to PrEP, which has
nothing to do with the current case. (Page 11: “About PrEP…” to “in practical terms”)
Answer to point 4: Thank you for the suggestion. We have removed the references to PREP in the text
and in the bibliography.
Point 5: The entire section from page 6 “In every case,…” up to page 7 “becomes more careful and
adherent” must be removed. It is at best a repeat of the introduction, and it must not be included in the
“case report” section.
Answer to point 5: We have removed the entire section, as suggested.
Point 6: Instead, the treatment history of the case should be included with specific drugs used duration
and resistance mutations (when available).
Answer to point 6: Thank you for the suggestion. We have added a new table (Table 2) which
summarises the patient's previous clinical history and the related available HIV resistance tests.
Point 7: Case report page 6 mentions a “60-year-old [person]” (not “patient”, please edit). Whereas the
case description page 8 indicates that this person is 70-year-old. Please edit.
Answer to point 7: Thank you for the suggestion. We corrected the sentence in the text. Sorry for the
mistake
Point 8: Please provide the GSS score for the OBR used. This is essential to interpret the results.
Without this information, statement regarding the success of LEN cannot be made (page 9). Notably, a
quick look at Stanford suggests no resistance against the NRTI combination and low-level resistance
against etravirine. There is also no baseline resistance against dolutegravir. Accordingly, it is likely that
the person would have achieved and maintained viral suppression even in the absence of IBA. Please
also discuss.
Answer to point 8: Following to the suggestion of the reviewer, we have added one figure to better clarify
the patient's resistance pattern and the remaining therapeutic options. (Fig. 1). The first part of figure
shows a resistance test performed in 2013, in which the presence of resistance mutations for INIs was
already evident, although these were not confirmed in subsequent tests. The second part of figure shows
the resistance test prior to the introduction of ibalizumab, with the corresponding Stanford interpretation,
which clarifies efficacy only for DTV and low-level resistance for the only other 2 available drugs. In this
case GSS for NRTI was 0,75 (potential low-level resistance), for NNRTI was 0 (high level resistance),
for PI was 0,03 (high level resistance), while no resistances were evident in RNA for INI. The third part
of figure shows a successive test performed on DNA in April 2022, that found archived resistances for
INI, as evidenced by the previous RNA test from 2013, with intermediate resistance only for DGT and
BIC. It is therefore clear that therapy with OBR alone could not control viral replication without nextgeneration drugs, and in fact, the patient had a viral load of 4,5 log.
Point 9: The authors must justify the choice of using IBA as monotherapy for 2 weeks before adding
the OBR; this is highly unusual.
Answer to point 9: Thank you for the suggestion. As per the ibalizumab protocol, an initial functional
monotherapy approach is used because it allows for the direct effect of this drug to be assessed without
interference from other antiretrovirals, observing the virologic response over a week. OBR is introduced
only after loading. These phases combine maximum antiretroviral activity and improved management of
multidrug resistance in patients with MDR virus. Typically, ineffective drugs are left in the functional
monotherapy phase, but in reality, this is no different from an initial true monotherapy. Furthermore, in
our case, the patient was already undergoing OBR therapy, which we would have reused in combination
with IBA. Therefore, we decided to temporarily suspend OBR and reintroduce it after the loading dose
of IBA, as per protocol. In this way we tried to optimize OBR effect after the initial reduction in viral
load induced by the monoclonal antibody.
Point 10: The report of disappearance of the resistance mutations in the proviral DNA may be one of
the interesting observations from this paper. However, clearance of resistance mutations by class have
been studied before. Please refer to these studies and discuss your results in this context. In addition,
can the authors report their target-not-detected/target detected results for this person? There exists a
possibility that the person is not fully suppressed with their OBR; thus, the authors should discuss the
possibility that they are not evaluating the “reservoirs” but rather viral replication in cryptic sites or
below the lower limit of detection of their assays. This must be discussed and mention of action on the
reservoir must be removed.
Answer to point 10: Thank you for the suggestion. The DNA resistance assessment was conducted for
research purposes, not due to a lack of efficacy in plasma of the current regimen, since the virus was
undetectable with salvage therapy. This demonstrated profound efficacy of the treatment, not related to
a viral value below the laboratory cut-off. Therefore, no residual viremia or replication in cryptic sites
was conceivable. Furthermore, our data are in accordance with other studies in the literature that show a
tendency for mutations stored in DNA to be cleared after persistent virological control. As suggested,
we have indicated the relevant literature data in the text. In our case, the progressive disappearance of
mutations in HIV-DNA, as evident in Fig. 5, is closely linked to the use of a fully effective salvage regimen
with IBA+OBR first and then with LEN+OBR. This is demonstrated by the fact that after stopping IBA
for 1 month, OBR alone, while still maintaining residual efficacy on plasma, was associated with the
reappearance of previously non-amplifying resistance mutations in the DNA. These would have
predisposed the patient to new virological failure in the plasma, if LEN was not introduced, associated
with OBR.
Point 11: Details about the DNA sequencing assay must be included in the Methods. Was this NGS,
Sanger? Has the method been published prior?
Answer to point 11: Thank you for the suggestion. We have specified in the methods that the Sanger
method was used. We have also added a bibliographic entry relating to the use of the Sanger vs DPU.
Point 12: Throughout the manuscript, please use first-person language that supports a respectful and
non-discriminatory way of talking about a person with HIV or others engaged in care.
Answer to point 12: Thank you for the suggestion. We corrected the sentences in the text.
Point 13: In the introduction, “destined to fail” is a too strong statement, please adjust.
Answer to point 13: Thank you for the suggestion. We corrected the sentence in the text with “was not
very efficacious”.
Point 14: Can the authors comment about the large difference in representation of HTE in the HIV
population comparing the US and Italy?
Answer to point 14: As suggested by the reviewer, we commented the statement in the text.
Point 15: In the introduction, page 3; please edit the possibly AI-generated statements: “Lower CD4
counts...” and “Co-infection with HCV…”, “multiple prior…”
Answer to point 15: Thank you for the suggestion. We corrected the sentence in the text.
Point 16: Page 3: “collateral” ïƒ side effects
Answer to point 16: We modified the sentence in the text according to the suggestion of the reviewer.
Point 17: Page 6: this section should be removed anyway but “incredible” is not a scientific term to
include in a publication
Answer to point 17: Thank you for the suggestion. The section had already been removed including the
term.
Point 18: “The problem with patients…” please edit; this is not an acceptable start of a sentence plus
the following statement concerns only those with resistance issues.
Answer to point 18: Thank you for the suggestion. We corrected the sentence in the text.
Point 19: In study design: “to evaluate… and the evaluation” please edit.
Answer to point 19: Thank you for the suggestion. We corrected the sentence in the text.
Point 20: Manufacturer’s information for the DNA Test PRO should be provided.
Answer to point 20: Thank you for the suggestion. We specified information in the text.
Point 21: Page 8: “His risk factor” is not first-person language appropriate. Please edit for something
like: “He reported engaging in unprotected intercourse with same-sex partners, which is a known risk
factor for HIV infection”
Answer to point 21: We corrected the sentence in the text, as suggested.
Point 22: Same consideration for Table 1 please “homosexual relationships”.
Answer to point 22: Thank you for the suggestion. We corrected Table 1 in the text.
Point 23: Page 9: “the pressure on reservoir”: unclear please edit and explain.
Answer to point 23: We corrected the sentence in the text with: “promoting the clearance of resistance
mutations stored in the reservoir, as evident from HIV-DNA analysis “.
Point 24: Page 9: “In this case, the drug has been”: very awkward, unclear statement; please edit.
Answer to point 24: Thank you for the suggestion. We corrected the sentence in the text making it
clearer.
Point 25: Some of the discussion repeats the introduction and should be removed to streamline the
discussion. The advantageous “posology” and potential benefits in terms of quality of life of LEN are
mentioned several times (3 or 4 times) throughout the manuscript. Please spare the reader and
streamline the text for readability.
Answer to point 25: Thanks for the suggestion. We corrected the text by removing the reference to the
benefits of "posology" from the introduction.
Point 26: Page 10: mentioning adherence improvement between IBA and LEN does not make sense
are both are administered.
Answer to point 26: Thanks for the suggestion. We referred to improved adherence to LEN, meaning
that administration was still performed every 6 months and not every 15 days as with IBA. After 12
months, the patient, in fact, was no longer able to continue treatment with IBA for logistical reasons.
We have, however, modified the text to reflect a possible improvement in adherence.
Point 27: Page 10: “On the other hand…”: stating that ART treatment is unrelated to CD4 counts is
incorrect. Please edit.
Answer to point 27: Thank you for the suggestion. We consider that recovery of the CD4 count is not
directly related to antiretroviral therapy, which only acts by blocking viral replication. We have,
however, modified the text adding that immunological recovery is not strictly related to the effect of
antiretroviral drugs.
Point 28: Page 11: “sequencing of [drugs]” does not exist. I assume the authors mean “sequential use
of […] drugs. Please edit.
Answer to point 28: Thank you for the suggestion. We corrected the sentence in the text.
Point 29: Page 9: “that highlighted mutations for all classes of drugs” is erroneous; no integrase
mutations could be detected. Please edit.
Answer to point 29: Thank you for the suggestion. We corrected the sentence in the text adding “all
classes except INSTI”.
We thank the Reviewer for helping us to improve our paper.
We hope that the paper is now worthy of publication in “IJMS”
Best regards,
Reviewer 2 Report
Comments and Suggestions for Authors
This is a rare report of treatment success with IBA and LEN, and it is considered to be a valuable report with novelty.
The clinical trials underlying the use of IBA and LEN are enough cited and discussed.
Drug resistance gene mutations were monitored over time, allowing for a detailed evaluation of the effects of IBA and LEN administration.
This paper is considered to be fundamentally worthy of publication.
The following comments aim to strengthen the manuscript and enhance its clinical and scientific impact.
Result
・Regarding side effects, how were the injection site reactions in this case?
・The improvement in quality of life is mentioned, but were any objective evaluations (e.g., HIVTSQc) conducted?
・Additionally, regarding medication adherence in the overall response (OBR), are there any objective indicators (e.g., Medication Adherence Report Scale)? If available, it would be appropriate to mention them.
Discussion
・The improvement in CD4+ cell counts is mentioned, which is encouraging as you pointed out. However, it is unclear whether this improvement is due to the recovery of bone marrow function or other factors that coincided with this period, or whether IBA or LEN played a role. It may be worthwhile to further discuss this, including previous reports.
・In the last part of the discussion, references 23-25 are cited, and PREP is mentioned. However, based on the content of this paper, the reference to PREP seems a bit of a leap. I think it is sufficient to mention that treatment was successful in patients with multidrug resistance.
・Needless to say, this is a case report, so it would be better to note that there is not yet sufficient data for generalization.
・Minor confirmation:
In the Introduction, there is “optimized background regimen (OBR),” followed by “low-efficacy residual therapy (OBR)” and “optimized residual backbone (OBR)” in the discussion. Does this mean that the terms in parentheses are abbreviations, or are they used in the sense of “that is”?
If so, no revision is necessary.
Author Response
ANSWERS TO THE COMMENTS OF THE REVIEWER 2
Point 1: Regarding side effects, how were the injection site reactions in this case?
Answer to point 1: Thank you for the suggestion. We added site reactions in the text.
Point 2: The improvement in quality of life is mentioned, but were any objective evaluations (e.g.,
HIVTSQc) conducted?
Answer to point 2: Thank you for the suggestion. No, we didn't consider it necessary to use specific
quality-of-life assessment tests. It was objectively clear that subcutaneous administration of a drug (LEN)
every 6 months was inevitably better accepted by the patient and guaranteed a better quality of life than
one requiring an intravenous infusion every 15 days (IBA). The improvement in quality of life has been
detected on anamnestic way. We clarified that in the text.
Point 3: Additionally, regarding medication adherence in the overall response (OBR), are there any
objective indicators (e.g., Medication Adherence Report Scale)? If available, it would be appropriate to
mention them.
Answer to point 3: No, we have not used any objective indicators.
Point 4: The improvement in CD4+ cell counts is mentioned, which is encouraging as you pointed
out. However, it is unclear whether this improvement is due to the recovery of bone marrow function
or other factors that coincided with this period, or whether IBA or LEN played a role. It may be
worthwhile to further discuss this, including previous reports.
Answer to point 4: The patient had previously had good virological control without ever achieving a
significant CD4 recovery. The fact that a salvage regimen with new-generation drugs was able to not
only provide virological efficacy but also promote a previously undeniable CD4 recovery suggests that
the new drugs indirectly played a role in this outcome. However, there is no data in the literature to
support this observation.
Point 5: In the last part of the discussion, references 23-25 are cited, and PREP is mentioned.
However, based on the content of this paper, the reference to PREP seems a bit of a leap. I think it is
sufficient to mention that treatment was successful in patients with multidrug resistance.
Answer to point 5: As suggested by the reviewer, we modified the text and the references eliminating
every mention to PREP.
Point 6: Needless to say, this is a case report, so it would be better to note that there is not yet
sufficient data for generalization.
Answer to point 6: We agree with the reviewer. Thus, in the conclusion we wrote that “new studies on
a larger population of patients with these characteristics and with longer follow-up will be able to
provide more data in this sense”.
Point 7: In the Introduction, there is “optimized background regimen (OBR),” followed by “lowefficacy residual therapy (OBR)” and “optimized residual backbone (OBR)” in the discussion. Does
this mean that the terms in parentheses are abbreviations, or are they used in the sense of “that is”?
If so, no revision is necessary.
Answer to point 7: Thank you for the suggestion. We used only OBR as an acronym in brackets,
writing out the full sentence that referred to OBR and not leaving it in brackets to avoid confusion in
interpreting the text.
We thank the Reviewer for helping us to improve our paper.
We hope that the paper is now worthy of publication in “IJMS”
Best regards,
Round 2
Reviewer 1 Report
Comments and Suggestions for Authors
Thank you for being so positively responsive to my comments.